# Microplastics in Fish and Fishery Products and Risks for Human Health: A Review

**DOI:** 10.3390/ijerph20010789

**Published:** 2022-12-31

**Authors:** Leonardo Alberghini, Alessandro Truant, Serena Santonicola, Giampaolo Colavita, Valerio Giaccone

**Affiliations:** 1Department of Animal Medicine, Productions and Health, University of Padova, Viale dell’Università 16, 35020 Legnaro, Italy; 2Department of Medicine and Health Sciences, University of Molise, 86100 Campobasso, Italy

**Keywords:** microplastics, fishery products, human health, environment

## Abstract

In recent years, plastic waste has become a universally significant environmental problem. Ingestion of food and water contaminated with microplastics is the main route of human exposure. Fishery products are an important source of microplastics in the human diet. Once ingested, microplastics reach the gastrointestinal tract and can be absorbed causing oxidative stress, cytotoxicity, and translocation to other tissues. Furthermore, microplastics can release chemical substances (organic and inorganic) present in their matrix or previously absorbed from the environment and act as carriers of microorganisms. Additives present in microplastics such as polybrominated diphenyl ethers (PBDE), bisphenol A (BPA), nonylphenol (NP), octylphenol (OP), and potentially toxic elements can be harmful for humans. However, to date, the data we have are not sufficient to perform a reliable assessment of the risks to human health. Further studies on the toxicokinetics and toxicity of microplastics in humans are needed.

## 1. Introduction

Over the past seventy years there has been a sustained increase in plastic production, rising from 1.5 million tons (Mt) in the 1950s to 367 million tons in 2020 [1]. The production of plastic and the consequent production of waste are related to the growth of the human population, which has increased from about 3.1 billion in 1961 to about 7.3 billion in 2015 and is projected to exceed 9 billion by 2050 [2]. The demands of this growth will drive the plastics commodity market, as will the demand for fishery and aquaculture products.

Estimates based on current growth rates indicate that plastic production is expected to double by 2025 and more than triple by 2050 [3]. The mass production and consumption of plastics have led to the accumulation of these materials in natural habitats resulting in negative impacts on biota and the economy [4].

Among the approximately 2.5 billion tons of solid waste produced globally in 2010, approximately 275 million tons were plastic waste generated and improperly managed by coastal countries and it is estimated that between 4.8 Mt and 12.7 Mt of this plastic waste has entered the oceans [5].

Once in the environment, plastic objects degrade and give rise to smaller fragments, which can directly enter the food chain or indirectly contaminate it due to the leaching of their potentially harmful chemicals [6].

Most humans ingest a significant amount of microplastic and even nanoplastic particles through food, particularly through the consumption of fish and other seafood [4]. To date, microplastics have been found in various foods such as beer, drinking water, honey, seafood, bivalve mollusks, sugar, and cooking salt [7,8].

To try to stem the problem of plastic pollution, many countries have set goals for the elimination or reduction of certain products such as plastic bags and disposable items [9].

In particular, in May 2018, a new proposal was approved by the European Commission, which bans several single-use plastic items and imposes stricter regulations on others [10]. Currently, 60 countries have banned or taxed the use of single-use plastics [11].

### 1.1. Management of Plastic Waste

Worldwide, the production of plastic waste is estimated to be 381 Mt per year and is expected to double in the next 10 years [12]. Only 9% and 12% of plastic waste was recycled and burned, respectively, while the remaining 79% ends up in landfills [13].

Waste treatment differs from country to country: developing countries are heavily dependent on landfills, while others focus on recycling and producing energy (in the form of heat, steam, and electricity).

To date, in the European Union, for example, 31% of plastic waste is destined for landfills [10]. However, not all plastics are recyclable or recycled and, if plastic waste is handled inappropriately, it can escape from waste management streams, enter the environment, and eventually reach the sea [14].

In general, plastic waste accounts for over 70% of all solid waste present in the natural environment [5]. The most recent data indicate that in 2020, mismanaged plastic globally was around 100 Mt and forecasts for the future suggest that it will reach around 220 Mt in 2060 if preventive actions are not taken to minimize the dispersion of plastic [15].

When dispersed into the environment, plastic waste is exposed to the intermittent combination of solar radiation, cooling, heating, drying, and rain, and begins to degrade and fragment, creating the so-called microplastics [16]. Over the years, the slow rate of plastic degradation has led to a significant accumulation of plastic debris on the surface of the sea, on the seabed, and on coasts around the world [17,18].

Current studies estimate that 88% of the sea surface is contaminated with plastic waste [19], of which 80% derives from terrestrial sources [20]. It has recently been estimated that 15 Mt of plastic enter the oceans each year and it has been hypothesized that these emissions of plastic waste will increase to around 20–53 Mt by 2030, potentially even reaching 90 Mt [21].

### 1.2. The Problem of Marine Waste

Marine litter is defined as any solid waste of human origin discarded at sea or which reaches the sea by dispersion from other environments [22,23]. They can be divided into nano-, micro-, meso-, and macro-wastes based on their size, and further divided into terrestrial and oceanic wastes based on their sources [24].

It is estimated that 40–80% of marine litter is made up of plastic [25]. Due to the high durability of plastic, such waste, reported all over the world, is destined to remain in the marine environment for a long period of time [26].

Household, river, marine, and fishing waste are generally considered chronic sources, as they represent an almost continuous input. These chronic sources combined with extreme weather events contribute to a large amount of plastic waste entering the marine environment [27].

Inevitably, in the marine environment, plastics undergo processes of mechanical abrasion, photodegradation, and biodegradation with the consequent formation of micro- and nanoplastics [28,29]. Plastic debris in the oceans varies widely in size, shape, and chemical composition [30].

## 2. The Microplastics

Although the presence of small plastic particles in the environment was already known since the early seventies [31], it is only recently that microplastics have attracted increasing interest from the scientific community.

In particular, in the last decade, the attention to the problem of microplastics as a new pollutant has seen a great increase in investments in this new field of research on a global level [32]. These plastic fragments cause many concerns as they are small enough to be ingested by living organisms and therefore reach humans through the consumption of contaminated food [33,34] in which the presence of microplastics was only reported at the beginning of 2010 [35].

### Definitions

To date, there is still no clear consensus on a definition broad enough to encompass all the criteria needed to describe microplastics [32]. An initial classification takes into consideration the dimensions of the fragments themselves, which can vary from several meters to such small dimensions as not to be visible to the naked eye (see Table 1) [36].

So far, the most widely used definition is that microplastics are particles smaller than 5 mm in their longest dimension. This definition has been adopted in practical terms as it considers the size below which ingestion by many species of marine biota occurs [38] and has been accepted by the NOAA (National Oceanographic and Atmospheric Administration) of the USA and by the MSFD (Marine Strategy Framework Directive) of the EU [14].

In the 2016 EFSA (the European Food Safety Authority) report, microplastics are defined as a heterogeneous mixture of materials of different shapes (fragments, fibers, spheroids, granules, pellets, splinters, or beads) whose dimensions are between 0.1 μm and 5000 μm, while nanoplastics are defined as plastic particles ranging from 0.001 μm to 0.1 μm [33].

Furthermore, microplastics are classified into primary and secondary. Primary microplastics are plastic particles intentionally produced in a size range of less than 5 mm and intended for special domestic or industrial uses [38].

Secondary microplastics, on the other hand, are the product of the fragmentation and degradation, caused by atmospheric agents, of larger plastics disseminated in the environment [38]. They appear to be the prevalent form in the marine environment, where it is estimated that approximately 68,500–275,000 tons of secondary microplastics are emitted annually [33]. Furthermore, it has been estimated that they constitute more than 90% of the 5 trillion microplastics floating on the sea [37].

## 3. Prevalence and Distribution of Microplastics in Fishery Products

According to FAO (the Food and Agricultural Organization of the United Nations) data, world production of fishery products reached about 179 million tons in 2018, of which 82 million tons came from aquaculture. Of the combined total, 156 million tons were used for human consumption, equivalent to an estimated annual supply of 20.5 kg per capita. The remaining 22 million tons were destined for non-food uses, mainly for the production of fishmeal and oil [39].

In the last decade, numerous articles have been published concerning the presence of microplastics in food, including fishery products (see Table 2) and sea salt [40,41,42,43,44,45,46,47,48,49,50,51,52,53,54,55,56,57,58,59,60,61,62].

The ingestion of microplastics has been observed in many species of fish intended for human consumption from the Pacific, Atlantic, and Indian Oceans, and the Mediterranean Sea, although only one or two microplastic particles have been detected per fish [23,63]. For example, microplastics have been observed in chub mackerel (*Scomber japonicus*), herring (*Clupea harengus*), mackerel (*Scomber scombrus*), Japanese anchovy (*Engraulis japonicus*), northern cod (*Gadus morhua*), blue whiting (*Micromesistius poutassou*), sprat (*Sprattus sprattus*), king mackerel (*Scomberomorus cavalla*), shortfin scad (*Decapterus macrosoma*), horse mackerel (*Trachurus trachurus*), hake (*Merluccius merluccius*), bream (*Pagellus acarne*), and common sole (*Solea solea*) [64,65,66,67,68,69,70,71,72,73].

Microplastics have also been found in migratory commercial fish species (e.g., *Thunnus thynnus*), seasonal migratory fish (e.g., sea bass and *Dicentrarchus labrax*) and sedentary fish (e.g., plaice and *Pleuronectes platessa*) [34] and in Mediterranean fish of great commercial importance [74] such as sardines (*Sardina pilchardus*) and anchovies (*Engraulis encrasicolus*) which are often consumed entirely [34]. In these studies, microplastics were identified predominantly in the digestive tract and rarely in edible tissues, such as muscles [40].

Given that most fish species are eviscerated before human consumption, direct human exposure to microplastics would be negligible [14]. However, evisceration does not necessarily eliminate the risk of human intake of microplastics. In fact, in a study conducted on two species of dried fish among the most commonly consumed (the green back mullet *Chelon subviridis* and sin croaker *Johnius belangerii*), it was found that the presence of microplastics was significantly greater in gutted fish (whole fish excluding viscera and gills) compared to excised organs (viscera and gills) [75].

Despite this, small pelagic fish such as sardines, herring, and other small freshwater fish are commonly eaten whole, posing a greater threat than gutted fish [76]. However, little is known about microplastic levels in small fish.

### 3.1. Bivalve Mollusks

Bivalve mollusks can filter and retain microplastics of various sizes in quantities that depend on their concentration and distribution in the marine environment.

Studies have revealed the presence of microplastics in two species of mussel commonly consumed as food by humans (*Mytilus edulis* and *M. galloprovincialis*) from five European countries (France, Italy, Denmark, Spain, and the Netherlands) [77]. In a study conducted on wild and commercial Belgian mussels (*M. edulis*, *M. galloprovincialis*, and *M. edulis*/*galloprovincialis* hybrid form), the number of total microplastics ranged from 0.26 to 0.51 per gram of mussels [78].

In samples of mussels (*M. edulis*) reared in the North Sea and of Pacific oysters (*Crassostrea gigas*) reared in the Atlantic Ocean, an average content of 0.36 and 0.47 particles/g, respectively, was detected; the quantities of particles were reduced to 0.24 and 0.35 particles/g, respectively, after a purification period of 3 days [79].

Another study counted, on average, 0.2 ± 0.3 particles/g with up to 1.1 particles/g of mussels collected in six locations along the coast of France, Belgium, and Holland and left for purification for 24 h. However, this study, having excluded fibers from the count, may have underestimated the concentration of microplastics [59].

In a study carried out on mussels (*M. edulis*) sampled in coastal waters and in UK supermarkets, Li and co-authors [80] found that in wild mussels the total microplastics, ranging from 0.7 to 2.9 particles/g of tissue and from 1.1 to 6.4 particles/specimen, exceeded those contained in farmed mussels purchased at the supermarket. Furthermore, in supermarket-bought mussels, the abundance of microplastics was significantly greater in pre-cooked mussels with 1.4 particles/g compared to fresh mussels with 0.9 particles/g [80]. These results suggested a possible contamination during processing rather than a real environmental contamination [81]. 

In contrast, higher quantities of microplastics were found in nine species of bivalves purchased in Chinese markets than in wild bivalves. In this study, in fact, the authors counted from 2.1 to 10.5 particles/g and from 4 to 57 elements per single specimen coming from the markets [82].

To date, the studies that have investigated the prevalence of microplastics in bivalve mollusks sampled in nature or on farms are numerous, while those on mollusks sampled in markets and supermarkets are still very few [72,81,82,83].

### 3.2. Crustaceans

The ingestion of microplastics in crustaceans can be both accidental and related to their dietary strategies [84]. Swimming crustaceans can ingest more microplastics than the sessile species. Crustaceans can be filter feeders, such as copepods and shrimps, opportunistic feeders such as the brown shrimp (*Crangon crangon*), or active hunters of small fish and other organisms, such as crabs and lobsters [85].

In a study conducted on brown shrimp (*Crangon crangon*) sampled from various locations in the English Channel, synthetic fibers were detected in 63% of the samples with an average value of 0.68 ± 0.55 particles/g (1.23 ± 0.99 particles/shrimp). In addition, seasonal differences have been reported, with greater absorption of microplastics in October compared to March [54]. Furthermore, in the same study, when comparing whole and peeled shrimp, no particles were detected in the abdominal muscles, which usually represent the edible portion of the shrimp. In fact, by shelling and removing the cephalothorax of the shrimp, the intestinal tract is removed, which contains most of the microplastics.

However, it is not always possible to remove completely the digestive tract from this small shrimp, thus exposing humans to possible ingestion of microplastics. For this reason, the author proposed to reduce the maximum number of microplastics that can potentially be eliminated by shelling and removing the cephalothorax to 90% [54].

In another study on Indo-Pacific shrimp (*Penaeus semisulcatus*) sampled in the Persian Gulf, the authors observed an average of 7.8 particles/crustacean and 1.5 particles/g, indicating the presence of microplastics both on the exoskeleton and in the muscle mass [86]. Other commercially relevant species in which microplastics have been observed include the Chinese mitten crab *Eriocheir sinensis* [87], the shore crab *Carcinus maenas* [88], and the Norway lobster *Nephrops norvegicus* [89].

Here, it is useful to remember that brown shrimps are an important food for a wide range of predators, thus facilitating the potential trophic transfer and accumulation of microplastics at the top of the food chain.

## 4. Risks for Human Health Related to the Absorption of Microplastics

While microplastic contamination is not a new phenomenon, the characterization of risks associated with micro- and nanoplastics is an emerging problem.

Risk is generally defined as the product of the hazard represented by an agent multiplied by the sustained exposure. The dangers posed by micro- and nanoplastics include physical characteristics (e.g., size, shape, and surface), additives used intentionally in the plastic manufacturing process, environmental durability, the ability to absorb chemical contaminants and pathogens and to concentrate them along the food chain [14].

Fishery products are known to be an important source of microplastics in the human diet. Therefore, if contaminated, their consumption can pose a threat to human health [90,91] (see also Figure 1).

Considering the recommendations of the EFSA, EUMOFA (European Market Observatory for Fisheries and Aquaculture Products), and NOAA for the consumption of fish by humans at different ages, the intake of microplastics, calculated on the ingestion of three species of fish that are caught and commonly consumed (*Dicentrachus labrax*, *Trachurus*, and *Scomber colias*), varies from 112 to 842 particles/year according to the EFSA and from 518 to 3078 particles/year/per capita according to the EUMOFA and NOAA [92].

Furthermore, it is relevant to consider that the degree of microplastic pollution and the consumption of bivalves varies considerably from country to country, leading to different levels of microplastic intake per capita in different countries each year [79,81,82,83]. For example, in European countries with high consumption of bivalves, it has been estimated that consumers ingest up to 11,000 particles/year/per capita, while in European countries with low consumption of bivalves, 1800 particles/year/per capita are ingested on average, which still represents a considerable exposure [79].

In South Korea, on the other hand, this estimate is 283 particles/year/per capita [83]. Furthermore, another study predicted that the consumption of mussels may lead to the ingestion of 123 particles/year/per capita in the UK and up to 4620 particles/year/per capita in the countries that consume the most bivalves (e.g., Spain, France, and Belgium) [56].

In addition, considering the average annual consumption of shrimp in Belgium (0.5 kg/person) and the best-case scenario where 90% of the microplastics are removed by shelling and cephalothorax removal, a quick indicative assessment revealed an ingestion rate between 15 and 175 particles/year for each person [54].

However, the information we currently have is not sufficient to assess the actual amount of microplastic to which humans are exposed, which represents a fundamental parameter for assessing the effects on human health [93].

### 4.1. Toxicity of Microplastics

Microplastics are considered potentially harmful to organisms depending on the degree of exposure and the susceptibility of the individual. Indeed, they can cause oxidative stress, cytotoxicity, and translocation to other tissues [94]. In organisms subjected to prolonged exposure, however, microplastics can lead to chronic flogosis, cell proliferation, necrosis, and impairment of immune cells [93]. In addition, microplastics can release chemicals (organic and inorganic) present in their matrix or previously absorbed by the environment [95] and act as vectors of microorganisms [96].

#### 4.1.1. Oxidative Stress and Cytotoxicity

Microplastics can cause oxidative stress due to the release of oxidizing species (for example, metals) that were previously adsorbed on their surfaces and reactive oxygen species (ROS) generated during the inflammatory response [97].

They also contain ROS produced by the polymerization and manufacturing process of plastics [98]. Oxidative stress after exposure to microplastics has been observed in mice [99] and zebrafish (*Danio rerio*) [100]. In humans, however, it has been reported that polypropylene (PP) prostheses induce an inflammatory response with the production of ROS that could lead to rejection [101].

Cytotoxicity is the result of particle toxicity, oxidative stress, and inflammation. Several studies have shown that exposure to microplastics and nanoplastics can potentially cause cytotoxicity. For example, an in vitro study revealed that polystyrene (PS) particles ranging in size from 20 to 40 nm are capable of producing cytotoxic effects on the Caco-2 cell line [102].

However, other in vitro studies have not reported large cytotoxic effects even using high particle concentrations (100–200 μg/mL) [103,104].

#### 4.1.2. Metabolic Disorders and Energy Balance

In humans, microplastics could have metabolic effects similar to those observed on mice and marine organisms. They can increase or decrease energy expenditure, reduce nutrient intake, and/or modulate metabolic enzymes. However, in humans the observation of these effects may be limited considering the low exposure concentrations, the greater energy requirements, and the greater complexity of human metabolism compared to the organisms tested [98].

#### 4.1.3. Translocation of Microplastics to the Circulatory System and Distant Tissues

After exposure, microplastics can act locally in the intestine or move to other organs through the circulatory system. Once in blood vessels, micro- and nanoplastics can cause a systemic inflammatory response, blood cell cytotoxicity by internalization, pulmonary hypertension, flogosis, and vascular occlusions [105,106].

The translocation of microplastics has been reported in rats, in which the particles, after ingestion and inhalation, reached the liver and spleen via the circulatory system [107]. In another study in mice, fluorescent polystyrene particles with a diameter of 5 μm and 20 μm accumulated in the liver, kidneys, and intestines [99].

Recently, microplastics ranging in size from 5 μm to 10 μm have been detected for the first time in the human placenta, in all its components: maternal, fetal, and amniochorial membranes [108]. Other in vitro studies have shown that PS nanoparticles (44 nm) can accumulate in human renal cortex epithelial cells with no cell effects and no signs of clearance in the first 90 min. However, the continued accumulation of these particles could impair renal function [109].

Furthermore, when transported to distant organs, microplastics can cause chronic inflammation, decreased organ function, and an increased risk of cancer [94].

#### 4.1.4. Disorders of the Immune System

After exposure, microplastics can induce local or systemic immune responses depending on their diffusion and the host response [98]. In particular, in genetically susceptible individuals, even just environmental exposure to microplastics is sufficient to interrupt the immune function causing autoimmune diseases or immunosuppression [94]. 

Furthermore, some studies have found that mussels exposed to microplastics exhibit immunosuppression and modulation of the immune response [105]. Although it has not yet been proven, microplastics could affect the immune system in humans as well. Therefore, further investigations of the effects on the human immune system are needed.

#### 4.1.5. Neurotoxicity

In vivo toxicity studies have shown that microplastics can affect neuronal function and behavior. In particular, in the brain of sea bass (*Dicentrarchus labrax*), it has been observed that these particles can cause inhibition of acetylcholinesterase, oxidative stress with increased levels of lipid peroxidation, and induction of anaerobic energy production pathways [76]. The same species also showed a negative impact on swimming performance, which is considered a behavioral indicator [110]. In mice, however, exposure to microplastics debris of polystyrene caused an increase in the activity of acetylcholinesterase and the serum levels of some metabolites that act as neurotransmitters (threonine, aspartate, and taurine) [99].

At present, however, very little is known about how microplastics could be involved in human neurotoxicity [94].

### 4.2. Effects of Contaminants and Additives Associated with Microplastics

Much of the concern raised by microplastics is due to the effects that could occur with the desorption of contaminants absorbed by the microplastics themselves and the chemical additives present in them [111].

In fact, plastic can concentrate contaminants up to the order of 106 [112]. Such contaminants could be released when microplastics come into contact with body surfaces and absorbed, thus reaching the underlying tissues [113].

#### 4.2.1. Plastic Additives

On average, 4% of the weight of the plastic produced is made up of additives [33]. However, the percentage of additives can vary significantly, in some cases even reaching half of the total material [114].

Teuten and co-authors [115] conducted a study on the amounts of additives present in microplastics collected from deep waters, coastal waters, and beaches. The results obtained by the authors can be summarized as follows: polybrominated diphenyl ethers (PBDE) from 0.03 nanograms per gram (ng/g) to 50 ng/g, bisphenol A (BPA) from 5 ng/g to 200 ng/g, nonylphenol (NP) from 20 ng/g to 2500 ng/g, and octylphenol (OP) from 0.3 ng/g to 50 ng/g.

Recently, research has also associated endocrine disruptors with various diseases and conditions, including hormonal cancers (breast, prostate, and testicles), reproductive problems (genital malformations and infertility), metabolic disorders (diabetes and obesity), asthma, and neurodevelopmental disorders (learning disorders and autism) [106].

An estimate of the exposure to additives by ingesting microplastics can be obtained as follows. Taking Chinese mussels (*Scapharca subcrenata* and *Alectryonella plicatula*) as an example, which contain on average the highest number of microplastics (4 particles/g) [82] and considering that an adult man generally consumes an average of 225 g of shelled mussels per serving, this meal would lead to ingestion of 900 microplastics. Assuming that these particles are spherical with an average diameter of 25 μm and a density of 0.92 g/cm^3^ (the density of low-density polyethylene, LDPE), the most common type of polymer of microplastics [116], these 900 microplastics would represent 7 μg of plastic, 4% of which would be additives. 

Using BPA as an example, this portion of mussels would contain 0.28 μg BPA. In a conservative scenario, we could assume that BPA is completely released from the microplastic. The EFSA estimated that an average adult BPA exposure from food and non-food sources of 0.19–0.20 μg/kg b.wt. per day [117]. Therefore, a 70 kg adult man would ingest about 14 μg of BPA per day on average. Consequently, bisphenol A from the microplastics of mussels would contribute about 2%, thus representing only a small part of the average daily exposure. 

#### 4.2.2. Bioaccumulative and Persistent Toxic Compounds (PBTs)

In addition to chemical additives, microplastics in the oceans can also accumulate persistent organic pollutants (POPs), including polychlorinated biphenyls (PCBs), polycyclic aromatic hydrocarbons (PAHs), and organochlorine pesticides such as DDT [71,112]. Due to their hydrophobic properties, POPs accumulate consistently in microplastics present in the marine environment. In fact, high concentrations of the following pollutants have been found in plastic particles taken from oceans, coastal areas, and beaches: non-dioxin-like polychlorinated biphenyls (PCBs) from 0.01 ng/g to 2970 ng/g; polycyclic aromatic hydrocarbons (PAHs) from 4 ng/g to 44,800 ng/g; and DDT and its analogues (DDD and DDE) from 2 ng/g to 2100 ng/g [115,118].

Once absorbed, POPs accumulate in human adipose tissue and their persistent characteristic makes them a serious threat to human health. In fact, exposure to these pollutants is associated with serious health problems such as endocrine disorders, reproductive problems, cancer, cardiovascular disease, obesity, and diabetes. 

Furthermore, prenatal exposure to POPs not only has negative effects on the health of the mother, but also on the newborn. Indeed, recent studies have shown that prenatal exposure to POPs may be associated with decreased birth weight [119], childhood obesity, increased blood pressure [120], and endocrine disrupting effects [121,122].

An estimate of exposure to POPs through the ingestion of microplastics can be obtained along the lines of that shown for plastic additives, using data on the presence of microplastics in bivalves from Li et al. [82], the amount of microplastics ingested from the consumption of 225 g of Chinese mussels (*Scapharca subcrenata* and *Alectryonella plicatula*), and data on the concentration of POPs in microplastics. 

In a conservative scenario, the highest organic pollutant concentrations measured in microplastics on a global scale would give a forecast of the maximum expected exposure. The highest POPs concentrations were found in microplastics deposited on beaches: PCBs up to 2750 ng/g and PAHs up to 24,000 ng/g. In this conservative scenario, these microplastics would lead to the ingestion of about 19 pg of PCBs and 170 pg of PAHs [33].

In the latest evaluation of PCB monitoring in food and feed available on the European market [123], the EFSA estimated an average exposure to non-dioxin-like PCBs of 0.3–1.8 μg of PCBs per day (for a human weighing 70 kg).

Regarding PAHs, the EFSA estimated an average exposure of 3.8 μg per day for the average EU consumer [124]. Therefore, even assuming that PCBs and PAHs are completely released from microplastics, consumption of these mussels would only have a small effect on exposure to PCBs (<0.006% increase) and PAHs (<0.004% increase) [33].

#### 4.2.3. Potentially Toxic Elements

Potentially toxic elements (PTEs) present in microplastics include Zn, Sb, Al, Br, Cd, Cu, Hg, As, Sn, Pb, Ti, Co, Cr, Ba, and Mn. They can be used as additives (e.g., in dyes, flame retardants, fillers, and stabilizers) during the plastic manufacturing process or they can adsorb in high concentrations to microplastics from the marine environment and, through the food chain, be transferred to aquatic organisms [69,76,125,126,127] and to humans [85,127,128].

The following metal concentrations were found in the microplastics collected along the south-western coasts of England: a few ng/g of Cd, Ni, and Cr; 7.7 μg/g of Cu; 10.3 μg/g of Pb; 171 μg/g of Al; 290 μg/g of Zn; 308 μg/g of Mn; and 314 μg/g of Fe [129].

Another more recent study, also along the south-west coast of England, found 3390 μg/g of Cd and 5330 μg/g of Pb in microplastics [130].

PTE toxicity depends on many different factors such as dosage, route of exposure, and chemical species, as well as the age, sex, genetics, and nutritional status of the exposed subject. A high concentration of PTEs in humans causes cell and tissue damage, leading to a variety of adverse effects and diseases [131,132,133,134].

A recent study has shown that mercury, lead, zinc, copper, and cadmium present on marine microplastics can lead to the co-selection of human pathogens resistant to antibiotics, posing a serious threat to humans when exposed to fishery products and marine environments [135]. 

Despite this, to date, no studies have been identified that have assessed the relevance of metals adsorbed to microplastics for food [33] and consequently for human health.

#### 4.2.4. Microbial Pathogens

Although it has been documented that microplastics can act as substrates for different microbial communities, there is insufficient data on the presence of such pathogens on microplastics to include them in risk profiling [14].

In summary, therefore, since the data we have available is very incomplete or lacking, it is not possible to carry out a reliable risk analysis to estimate the effects of microplastics on human health. However, with regard to contaminants and additives, the estimated exposure through the ingestion of the microplastics contained in fishery products constitutes only a fraction of the total daily dietary intake of these compounds.

Furthermore, considering that the diet is not the only source of contaminants and additives, this exposure is overestimated. It can therefore be concluded that the effect of the consumption of microplastics on the intake of contaminants and additives is most likely negligible [14].

## 5. Conclusions

Mass consumption, inadequate waste management, and the durability of plastics have led to a huge accumulation of plastic waste of all sizes in aquatic environments around the world. Current estimates predict that plastic production levels will continue to increase exponentially in the near future. As a result, related marine pollution is also likely to increase.

Microplastics have been detected in all aquatic compartments; hence, they are currently considered an omnipresent contaminant that constitutes a potential risk both for aquatic organisms that ingest them and for humans through the consumption of fishery products.

The ingestion of microplastics by aquatic organisms, including fish species of commercial interest to humans, has been confirmed by laboratory and field studies.

The quantities of microplastics observed from the analysis of the intestinal contents of these organisms are generally low. However, data on the presence of microplastics in some species of interest, including small fish, and in tissues outside the digestive tract are still very limited. Furthermore, it is not known whether the negative effects resulting from the ingestion of microplastics by aquatic organisms under experimental conditions may also occur in nature.

There are currently no guidelines for the analysis of microplastics and only limited methods are available for the identification and quantification of these particles. Furthermore, the methods described for the degradation of organic matter all have disadvantages, including the degradation to some extent of certain types of plastics.

It is now well documented that microplastics contain additives added during the plastic manufacturing process and that microparticles can efficiently absorb chemical and biological contaminants from the environment. Such contaminants can be transferred to organisms once the microplastics are ingested. 

However, considering the calculations are based on experimental studies, a critical evaluation of the literature related to the transfer of PBTs from microplastics to marine animals concluded that the ingestion of microplastics by marine organisms is not capable of increasing PBT exposure in these specimens. 

Similarly, the transfer of additives from ingested microplastics seems to have a negligible effect on total exposure to the additives, as has been demonstrated for NP and BPA.

According to the FAO, the per capita consumption of fish products has increased dramatically, from 9.0 kg in 1961 to 20.5 kg in 2018. However, the amount of microplastics ingested by humans through the consumption of fishery products is still poorly determined. As a rule, most fish species are eviscerated before consumption, greatly reducing the risk of ingestion since most of the particles are contained within the gastrointestinal tract. Exceptions to this rule are bivalve mollusks, small crustaceans, and some species of small fish, which are generally eaten whole.

In humans, various potential toxic effects of microplastics have been hypothesized; however, to date, the data we have is not sufficient to make a reliable assessment of the risks to human health. Although it is currently believed that the overall risks to human health from ingesting microplastics present in fishery products are low, it is important to consider that microplastics present in aquatic environments will inevitably increase, due to the degradation of the already present plastic and plastic that will enter in the future.

In the face of all this, it is important and urgent that researchers carry out further studies to fill the current gaps in the knowledge of microplastics. In addition, analytical methods for the detection and quantification of microplastics in aquatic environments and in fishery products should be further developed, standardized, and validated, with a focus on particles smaller than 150 μm.

Furthermore, to make a reliable assessment of human exposure to microplastics it is important to acquire further data on the presence and size of these particles in fishery products, and to understand the effects that food processing could have on microplastics.

Finally, further research is needed on the toxicokinetics and toxicodynamic of microplastics to both aquatic organisms and humans, including local effects on the intestinal tract, interactions with the microbiota, and the potential formation of nanoplastics resulting from the degradation of microplastics in the gastrointestinal tract. In this context, more studies must be carried out on microplastics as vectors of pathogens for fishery products and for humans.

## Figures and Tables

**Figure 1 ijerph-20-00789-f001:**
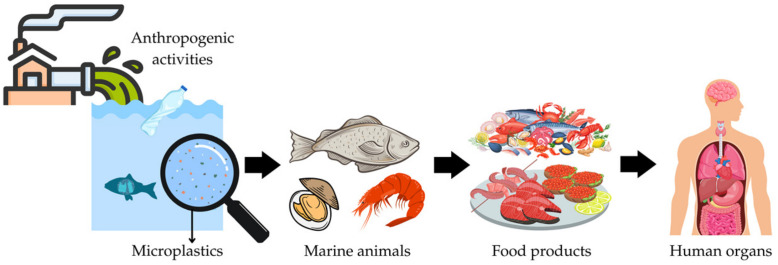
A model showing how anthropogenic activity cause microplastics to enter the food web, make a path to our food and, ultimately, into our organs.

**Table 1 ijerph-20-00789-t001:** Size categories of plastic marine litter, assuming a near-spherical form, showing common definitions and alternative options that may be appropriate for operational reasons from [36].

Field Descriptor	Common Size Divisions	Measurement Units	Alternative Options	Remarks
Mega	>1 m	meters		
Macro	25–1000 mm	meters centimeters millimeters	25–50 mm 1–25 mm	
Meso	5–25 mm	centimeters millimeters	<25 mm	MARPOL Annex V (pre-revision)
Micro	<5 mm	millimeters microns	1–5 mm <1 mm >330 μm	[37]
Nano	<1 μm	nanometers	<100 nm	Not considered for monitoring

**Table 2 ijerph-20-00789-t002:** Microplastics in fishery products.

Species	Number of MPs Detected	% of Samples Positive to MPs (*)	Sampling Site	Reference
**Fish**				
*Merluccius*	0.40 ± 0.89 ^a^	29%	Atlantic Ocean, Portugal	[41]
*Merluccius merluccius*	1.38 ^a^	26.8%	Mediterranean Sea	[42]
*Merluccius merluccius*	1.6 ± 0.6 ^a^	40%	Central Adriatic Sea, Italy	[43]
*Mullus barbatus*	1.08 ^a^	19.7%	Mediterranean Sea	[42]
*Mullus barbatus*	1.8 ± 0.24 ^a^	22.47%	Western Mediterranean Sea, Italy	[44]
*Mullus barbatus*	2 ^a^	20%	Central Adriatic Sea, Italy	[43]
*Engraulis encrasicolus*	1.25 ^a^	91%	Adriatic Sea, Italy	[45]
*Engraulis encrasicolus*	2.5 ± 0.3 ^a^	83.4%	Eastern Mediterranean Sea, Lebanon	[46]
*Engraulis encrasicolus*	0.12 ± 0.12 ^a^	34%	Eastern Ligurian Sea, Italy	[47]
*Sardinia pilchardus*	4.63 ^a^	96%	Adriatic Sea, Italy	[45]
*Scomber scombrus*	0.29–2.5 ^a^	100%	Atlantic Ocean, Portugal	[48]
*Mugil chephalus*	2.5 ^a^	13.8%	Hong Kong coast	[49]
*Mugil chephalus*	10 ± 9 ^a^	97%	Minho estuary, Iberian Peninsula	[50]
**Crustaceans**				
*Penaeus semisulcatus*	0.36 ^b^	64.1%	Persian Gulf, Iran	[51]
*Parapenaeopsis hardwickii*	0.25 ^b^	45%	East China Sea, China	[52]
*Fenneropenaeus indicus*	0.04 ^b^	30.9%	Kerala coast, India	[53]
*Portunus armatus*	0.26 ^b^	64.1%	Persian Gulf, Iran	[51]
*Crangon crangon*	0.55 ^b^	63%	Channel area and Southern North Sea between France, Belgium, the Netherlands, and the UK	[54]
Mixed crab species	0.53 ^b^	91%	Hong Kong coast	[55]
**Mollusks**				
*Mytilus* spp.	3.2 ± 0.52 ^c^	-	Edinburgh, UK	[56]
*Mytilus galloprovincialis*	0.8–0.9 ^c^ 2.5–5.3 ^d^	46.25%	Ionian Sea, Greece	[57]
*Mytilus galloprovincialis*	0.69 ^c^ 0.23 ^d^	48%	Turkish coasts	[58]
*Mytilus edulis*	0.61 ± 0.56 ^c^ 0.23 ± 0.20 ^d^	65–90%	Nantes, France	[59]
*Mytilus edulis*	0.76 ± 0.40 ^c^ 0.15 ± 0.06 ^d^	34–58%	Channel coastlines, France	[60]
*Crassostrea* spp.	2.93 ^c^ 0.62 ^d^	84%	coastline of China	[61]
*Crassostrea gigas*	2.10 ± 1.71 ^c^ 0.18 ± 0.16 ^d^	80–93%	Nantes, France	[59]
*Crassostrea gigas*	0.69–3 ^c^ 0.02–0.14 ^d^	63%	Salish Sea, USA	[62]
*Cerastoderma edule*	2.46 ± 1.16 ^c^ 0.74 ± 0.35 ^d^	34–58%	Channel coastlines, France	[60]

* In this context, the term “positive” expresses the number of fish samples, which, during the analysis, were found to be contaminated by MPs. ^a^ MP detected in the gastrointestinal tract. ^b^ MP content in the edible section. ^c^ MPs/individual. ^d^ MPs/g ww.

## Data Availability

Data sharing not applicable.

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
