# Peer review of "Microplastics in Fish and Fishery Products and Risks for Human Health: A Review"

_ijerph, 2022, doi:10.3390/ijerph20010789_

Round 1

Reviewer 1 Report

The risks of microplastics to aquatic organisms and human health are a globally concerned topic. The authors introduced the historical and current situation of plastic waste and the definition of microplastics. Furthermore, the authors reviewed the distribution of microplastics in fishery products and the potential risks to human health.

Major comment: For the Toxicity of microplastics section, it is important to articulate the levels of microplastics introducing a variety of toxicity effects (if possible).

Minor comments:
L18: "heavy metal": The term "heavy metal" has been widely used for decades, but more and more criticism has been raised against using this term, such as this reference below. I suggest considering a different term as appropriate throughout the manuscript.
Reference: Pourret, O.; Hursthouse, A. It's Time to Replace the Term "Heavy Metals" with "Potentially Toxic Elements" When Reporting Environmental Research. Int. J. Environ. Res. Public Heal. 2019, 16, 4446, doi:10.3390/ijerph16224446.

L36: The abbreviation of “Mt” has been defined on L26 already.

L104: Note that reference 35 was published in 2018. However, the authors stated the “the presence of microplastics was only reported at the beginning of 2010”. Please clarify. I would suggest citing the original publication if applied.

L114: I suggest adding a note stating which part was modified. Same comment for L250.

Table 1: Please add a line between Micro and Nano to keep consistency.

L122: Please define EFSA (European Food Safety Authority)

L137: Please define FAO.

Table 2: Please clarify what “positive” means here. Is that mean MPs interact positively with the effect of other contaminants? Or the presence of MPs in fishery products is positively correlated with MPs concentrations? Also, if possible, please keep decimals and Sample site descriptions consistent.

L236: I am unsure if the authors meant “assumption” or “consumption”.

L252: Please define EUMOFA. Same comment for LDPE on L371.

L291: I suppose PP stands for Polypropylene? Same comment for L318, which I think PS stands for Polystyrene.

L349: Please specify what contaminants.

L354: I suggest not using the term “significantly” here unless a statistical analysis was conducted.

L367 - 368: Please check the reference of this statement. From Reference 80, Microplastics in mussels sampled from coastal waters and supermarkets in the United Kingdom, the mussels seem unlikely a Chinese species or the mussel used in this literature (Mytilus edulis) is called “Chinese mussel” locally or the authors refer to Reference 81? If so, please refer scientific species name. Same comment on L404.

L381: This one-sentence paragraph seems too general. I would suggest the authors add some cautious wording.

L410: I suppose “PHAs” should be “PAHs”?

L424: I suggest changing “man” to “human”, unless the authors specify data from different genders.

L464: I agree that the accumulation of microplastics by aquatic organisms has been confirmed by lab and field studies, but it seems to lack mesocosm scale research, to some extent, which may lead to a potential for future research.

L507: The authors mentioned toxicokinetics as a potential future research, should toxicodynamic also be considered, rather than toxicity only?

L523: “]” should be “)”.

L546: I think the “e” between “isolating” and “identifying” should be “and”. Same comment for L706

L600: I suppose the “e”  between “fate” and “effects” should be “and”. It would be great to add access website, if it is the case. http://www.gesamp.org/site/assets/files/1272/reports-and-studies-no-90-en.pdf

Author Response

We are grateful to the Reviewer for his very helpful comments on our review. It is evident that he is an expert on the subject. We have collected our changes and our answers to his questions in the attached file. So, please, see attached file!

Reviewer 2 Report

This review talks about the recent environmental problem caused by plastics all over the world but is especially centred on fish and fishery products related to their risk to human health.

The authors develop a complete introduction that allows the reader to get the keys to understanding the roots of this huge problem. Plastics have become to be studied from different points of view and, in my opinion, due to their nature and the way to study them, there are many different forms to face it. This means that is necessary to simplify or unify nomenclature, units, analytical methods, etc. That is well gathered in table 1.

I think that the main species selected to carry out this review can be representative of the much-consumed ones. Even though, the authors also point out some others taking into account the great variety of habits in the world.

I also think that is important the way to expose the section of risks for human health. In my opinion, the subsections are all appropriate and well resumed. I would emphasize especially the approach made with some attempts at theoretical quantification from data of microplastics in fish to translate them into concentrations in humans and their possible evaluation.

Conclusions perfectly gather the main significant ideas along the text and the authors also propose some ways for future works. This can be obvious but in my opinion, the manuscript marks what is still to do around the immense field of plastics.

Anyway, I would add a few details to be changed or weighed up by the authors:

·    Abstract. L. 17-18. I think that a comma must be added to the compounds.

·    Table 1. Please, rewrite the last unit (Nanometeters) in the "Measurement units" and do not use a capital letter as in the other units too.

·    L. 118. 5 mm

·    Table 2. Capital letter for the sampling site Eastern Ligurian Sea.

·    L. 410. PAHs instead of PHAs.

·    In reference 7, the editors and DOI could be added to complete it.

Finally, I have a curiosity or question about the authors. As they conclude, there are no guidelines about many details related to the study of plastics (micro, nano, etc.) and their quantification. That is something evident when reading table 2 and comments on pages 5 and 6. Some different forms to deal with their quantification are presented in the references. For instance, particles/g mussels, particles/g tissue, particles/specimen, elements/single specimen. I understand the importance to unify and standardize units for every type of study. In this sense, I consider that in table 2, maybe would be interesting to lighten the meaning of MPs/individual columns. Some data are expressed as MPS/g (ww), others are percentages, and some others are MP content in the edible section. However, when no indications are given, what is the meaning?

Author Response

(The authors gave the same response as above.)

Reviewer 3 Report

MANUSCRIPT: 2122078

TITLE: Microplastics in Fish and Fishery Products and Risks for Human Health: A Review.

The manuscript 2122078 “Microplastics in Fish and Fishery Products and Risks for Human Health: A Review.”, presents a review of the literature.

The manuscript presented is well structured.

The review is clearly written, well systematized and comprehensive for the topic, and the literature cited is adequate and most of the papers cited are from the last five years, which demonstrates the relevance and interest in the topic of microplastics in health

Similar reviews are not known and it is of much interest to the scientific community.

The conclusions are consistent and in accordance with the quotes listed.

References are in accordance with the MDPI Reference List and Citations Style Guide

I congratulate the authors for this systematized review of high interest to the scientific community where they present a review that is very easy to read.

Author Response

In this case, Reviewer 3 did not specifically comment on our review, but appreciated and praised our work. We are extremely grateful for his esteem!